# Searching for Novel Biomarkers in Thymic Epithelial Tumors: Immunohistochemical Evaluation of Hippo Pathway Components in a Cohort of Thymic Epithelial Tumors

**DOI:** 10.3390/biomedicines11071876

**Published:** 2023-07-01

**Authors:** Kostas Palamaris, Georgia Levidou, Katerina Kordali, Christos Masaoutis, Dimitra Rontogianni, Stamatios Theocharis

**Affiliations:** 1First Department of Pathology, National and Kapodistrian University of Athens, 11527 Athens, Greece; kpalamaris@yahoo.gr (K.P.); katerinakordali@hotmail.com (K.K.); cmasaout@med.uoa.gr (C.M.); dgian@otenet.gr (D.R.); 2Department of Pathology, Paracelsus Medical University, 90419 Nuremberg, Germany

**Keywords:** thymic epithelial tumors, thymoma, thymic carcinoma, Hippo, YAP/TAZ, TEAD4, LATS1, Masaoka–Koga, biomarkers

## Abstract

Given the pivotal role of the Hippo pathway in different facets of tumorigenesis, which has been vigorously established in multiple heterogenous malignancies, we attempted to evaluate its potential utility as a prognostic–predictive biomarker in thymic epithelial tumors (TETs). For this purpose, we performed a comprehensive immunohistochemical analysis of four Hippo cascade components (YAP, TAZ, TEAD4 and LATS1) in a sizeable cohort of TETs and attempted to identify possible correlations of their H-score with various clinicopathological parameters. TAZ and TEAD4 displayed both cytoplasmic and nuclear immunoreactivity in almost equal frequency, with their cytoplasmic H-score being strongly associated with more aggressive high-grade tumors (type B3, thymic carcinoma) and more advanced pathological stages. On the other hand, a primarily nuclear staining pattern was encountered in both YAP and LATS1, with the YAP nuclear H-score being higher in more indolent (type A) and earlier stage tumors. Interestingly, none of the four examined factors displayed any statistically significant correlation with patient overall (OS) or disease-free survival (DFS). In summary, our results provide some initial insight into the expression profile of these core Hippo pathway components in thymic neoplasms and point towards some clear associations with tumor characteristics, which are of paramount translational-clinical research with profound implications in therapeutic targeting of this pathway in the context of precision medicine.

## 1. Introduction

The thymus gland is a major secondary immune organ, with an indispensable role in regulating T-cell maturation and differentiation. Histologically, it is segregated into two zones: medulla and cortex, composed of distinct epithelial cell populations. Epithelial cells of the gland are capable of forming neoplasms, with an epithelial phenotype, termed as thymic epithelial tumors (TETs). Between its two distinct histological compartments, medullary epithelial cells are believed to be the cells of origin of such neoplasms, even though this assumption is based only on very preliminary data [1]. Apparently, the occurrence of a tumor that occupies part of the thymus gland impairs the network of interactions among naïve T-lymphocytes and thymic epithelial cells and disrupts the selection steps [2,3]. It is, therefore, a normal consequence that such neoplasms are linked with a higher propensity for the development of systematic autoimmunity, with Myasthenia Gravis being the most frequent autoimmune entity encountered in patients with TETs [4,5]. The generally low prevalence of TETs renders the precise clarification of their pathogenesis a pretty tough challenge and, as expected, the exact molecular alterations that mark both the initiation and the progression of the disease have yet to be delineated. On the other hand, some recent in vivo experiments, on genetically engineered mouse models (GEMMs) based on tissue-specific ablation of different components of the Hippo pathway, have unraveled a crucial function of this signal transduction system, in both thymus gland development and in the maintenance of its histological architecture and functional integrity [6]. From a molecular perspective, Hippo is an evolutionary conserved signaling pathway, with a multidimensional role in regulating embryogenesis and homeostasis in various tissues [7,8]. Upon the binding of appropriate ligands on the respective transmembrane receptors, a cascade of intracellular phosphorylation events, mediated mainly via large tumor suppressor kinase 1/2(LATS1/2), an intracellular kinase, that ends up to two transcription co-factors, YAP and TAZ, resulting in their cytoplasmic shuttling, making them unable to exert their transcription-regulation functions. Thus, inhibition of the Hippo cascade attenuates this inhibitory effect on YAP/TAZ, allowing them to produce their transcription output [9]. Even though YAP and TAZ are legitimately considered as the undeniable key components of the Hippo signaling cascade, their lack of DNA-binding domains means that they cannot, by themselves, coordinate the reconstruction of a cell’s transcriptome. In order to compensate for this lack of direct gene-expression-modifying ability, they are equipped with specialized structural domains that allow them to recognize other conventional transcription factors, which belong foremost to the TEA Domain Transcription Factor (TEAD) family, with TEAD4 being the prototypical representative [10,11,12]. The latter binds to the TEA domain, forming heterodimers, which orchestrate a reprogramming of the gene expression and establish specific patterns across multiple cell lineages, modifying the differentiation of stem-like cells toward a variety of mature cell identities that help occupy and repopulate damaged tissue [13]. As expected, their eminent function in preserving tissue homeostasis postulates an equally important role in initiation and maintenance of oncogenesis. Indeed, a well-accepted role has already been attributed to the Hippo pathway in the emergence and evolution of multiple malignant tumors. YAP/TAZ have been found overexpressed in a diverse array of heterologous malignancies, and mechanistically, they have been associated with the acquirement of cancer hallmark features, as well as with the maintenance and function of cancer stem cells and the development of resistance to mainstream therapeutic schemes [14,15,16,17,18]. Thus, in this paper, we aimed to evaluate the expression patterns of YAP, TAZ, TEAD4 and LATS1 in a cohort of TETs, distributed across the whole breadth of their histological subclasses and correlate them with available clinicopathological parameters (Table 1). 

## 2. Results

### 2.1. Immunohistochemical Expression of YAP in TETs

YAP expression in epithelial cells is mainly nuclear and was detected in 76.9% of the investigated specimens, 32% of which also showcased a cytoplasmic immunoexpression (Figure 1, Table 2). The YAP nuclear and cytoplasmic H-score were positively associated (Spearman’s correlation coefficient R = 0.3723, *p* = 0.0128). Epithelial-rich TETs [B3, Thymic Carcinoma (TC)] showed a lower nuclear YAP H-score compared to the rest of the TETs (Mann–Whitney, *p* = 0.0042, median value 45 vs. 5, Figure 2A). TETs of advanced Masaoka–Koga stages (III/IV) showed a lower nuclear YAP H-score compared to the other types (Mann–Whitney, *p* = 0.042, median value 30 vs. 10, Figure 2B). The YAP nuclear H-score was negatively associated with the TAZ cytoplasmic H-score (Spearman’s correlation coefficient R = −0.4784, *p* = 0.0010). No significant correlation was identified between YAP expression and the remaining clinicopathological parameters, such as patients’ overall survival (OS) or disease-free survival (DFS) (*p* < 0.10).

### 2.2. Immunohistochemical Expression of TAZ in TETs

TAZ immunoreactivity in TETs was both cytoplasmic and nuclear. A positive staining was observed in 96.2% of the cases. A total of 74.7% of the cases displayed nuclear and 72.1% cytoplasmic immunopositivity (Figure 1, Table 2). Epithelial-rich TETs (B3, TC) showed a higher cytoplasmic and lower nuclear TAZ H-score compared to the other types (Mann–Whitney, *p* = 0.0001 for cytoplasmic, median value 200 vs. 55 *p* = 0.0215 for nuclear, median value 0 vs. 20, Figure 3A). TETs of advanced Masaoka–Koga stages (III/IV) demonstrated a higher cytoplasmic TAZ H-score compared to the other types (Mann–Whitney, *p* = 0.0308, median value 100 vs. 65, Figure 3B). No significant correlation was identified between the TAZ H-score and the remaining clinicopathological parameters, including patients’ OS and DFS (*p* < 0.10).

### 2.3. Immunohistochemical Expression of TEAD4 in TETs

TEAD4 staining was observed in 98.3% of the evaluated specimens, with 91.4% displaying cytoplasmic immunopositivity and 93% nuclear immunopositivity (Figure 1, Table 2). Fifty cases (86.2%) showed concurrent nuclear and cytoplasmic positive staining, whereas only one case did not display any positive staining, neither cytoplasmic nor nuclear. TEAD4 nuclear and cytoplasmic H-scores were positively associated (Spearman’s correlation coefficient R = 0.36, *p* = 0.017). Epithelial-rich TETs (B3, TC) showed a higher cytoplasmic TEAD4 H-score (Mann–Whitney, *p* = 0.002, median value 200 vs. 95, Figure 4A), while tumors of increased Masaoka–Koga stages (II/III/IV) displayed a marginally higher cytoplasmic TEAD4 H-score (Mann–Whitney, *p* = 0.05, median value 150 vs. 75, Figure 4B). The TEAD4 cytoplasmic H-score was positively correlated with the cytoplasmic LATS1 H-score (Spearman’s correlation coefficient R = 0.42, *p* = 0.005) and the cytoplasmic TAZ (Spearman’s correlation coefficient R = 0.30, *p* = 0.048), while it also demonstrated a negative correlation with the YAP nuclear H-score (Spearman’s correlation coefficient R = −0.41, *p* = 0.006). No significant correlation was identified between TEAD4 expression and the remaining clinicopathological parameters presented in Table 2, such as patients’ OS or DFS (*p* < 0.10).

### 2.4. Immunohistochemical Expression of LATS1 in TETs

Positive immunoreactivity for LATS1 was observed in 77.4% of the cases (Figure 1, Table 2). All the positive cases showcased nuclear immunoreactivity, 6 of them also showed a cytoplasmic staining. LATS1 nuclear and cytoplasmic H-scores were positively associated (Spearman’s correlation coefficient R = 0.3332, *p* = 0.0271). The epithelial-rich TETs (B3, TC) more often displayed a cytoplasmic LATS1 immunoexpression (Fischer’s exact test, *p* = 0.006, 27.7% versus 2.3%, Figure 5). The same applied to the more advanced stage tumors (III/IV), which were more often LATS1 cytoplasmic positive (Fischer’s exact test, *p* = 0.04, 25% versus 4%). In addition, a negative correlation was observed between the LATS1 cytoplasmic H-score and the TAZ nuclear H-score (Spearman’s correlation coefficient R = −0.3464 *p* = 0.0213). No significant association was identified between the LATS1 H-score and the remaining clinicopathological parameters, including patients’ OS or DFS (*p* < 0.10).

## 3. Discussion

Aberrant regulation of the Hippo pathway is a mainstream feature of numerous malignant neoplasms of divergent origins and differentiation, including glial tumors, epithelial carcinomas (colorectal, pancreatic, breast, lung, liver, prostate, squamous and bladder cancer), melanoma and sarcomas [19,20,21,22,23,24,25,26,27,28,29,30,31,32]. Given the lack of data regarding the role of Hippo in thymic tumorigenesis, we have made the first thorough immunohistochemical evaluation of four core factors of the pathway (YAP, TAZ, TEAD4 and LATS1) in a cohort of cases distributed among all the different TET histological subtypes. 

The two hallmark mediators of the Hippo cascade, YAP and TAZ, displayed interesting and dissimilar expression patterns. YAP was primarily found within the tumor cells’ nucleus, as 76.9% of our specimens demonstrated nuclear positivity. YAP nuclear staining was mainly encountered in lower-grade neoplasms, namely A, AB, B1 and B2 types, compared to the more malignant B3 and C tumors (*p* = 0.0042), while statistically significant differences of YAP immunoreactivity were also observed among different Masaoka–Koga stages, as early tumor stages (I–II) were associated with a higher H-score. Contrary to YAP, TAZ showcased immunoreactivity in almost all examined samples (positivity rate: 96.2%), and it was encountered in both the nuclear and cytoplasmic compartment of neoplastic cells with similar frequency (nuclear positivity: 74.7% and cytoplasmic positivity: 72.1%) but distinct staining distribution. Cytoplasmic TAZ immunoreactivity was higher in high-grade, epithelial-rich B3 and C tumors compared to the other subtypes (*p* = 0.0001), while nuclear staining demonstrated a reverse pattern, with higher protein levels in neoplasms with more indolent behavior (A, B1, B2) (*p* = 0.0215). An increased cytoplasmic H-score was also associated with more advanced pathological stages (III–IV) (*p* = 0.0308). TEAD4 reactivity was detected in 98.3% of our specimens, with the vast majority of them (86.2%) displaying bimodal nuclear and cytoplasmic staining. Cytoplasmic TEAD4 was significantly higher in epithelial-rich B3 and C neoplasms (*p* = 0.002) and in advanced Masaoka–Koga stages (*p* = 0.05). Finally, LATS1 was observed in both the nucleus and cytoplasm of neoplastic cells. Nuclear staining was detected in 77.4% of the examined samples, while cytoplasmic immunoreactivity was encountered in six cases of the epithelial-rich B3 and C types. Cytoplasmic positivity for LATS1 was also more frequent in advanced pathological stages. 

While staining patterns observed in our cohort are in large part consistent with findings from previous studies, there are still some deviations from the trends identified in other tumor subtypes. Regarding the expression of YAP/TAZ, we can conclude that nuclear YAP and TAZ are associated with less aggressive neoplasms, while cytoplasmic TAZ correlates to high-grade and more advanced tumors. TEAD4, which serves as the de facto transcriptional executor of the pathway, is also upregulated in more aggressive neoplasms and advanced pathological stages. The overexpression of YAP in lower-grade tumors reconciles with the findings derived from an integrated molecular analysis of a large TET cohort, which unveiled YAP as one of the upregulated genes in tumors harboring GTF2I mutations that represent the hallmark genetic driver of low-grade thymic tumors [33]. In multiple studies, YAP/TAZ and TEAD4 overexpression was detected in a diverse array of malignancies, engulfing lung and breast tumors, neoplasms of the gastrointestinal tract, extending from the esophageal to the colorectal region, as well as of the accessory glands (liver and pancreas), where it has been exclusively associated with detrimental prognosis and unfavorable clinicopathological parameters, such as higher tumor grade, advanced stages and resistance to treatment [34,35,36,37,38,39,40,41,42]. The unfavorable impact of YAP/TAZ overexpression in patients’ survival and response to therapeutic schemes has also been verified with complementary experimental studies conducted in cell lines and genetically engineered mouse models. A multiplicity of experiments, based on either artificially forced overexpression or genetic ablation/knock down of the two factors, have demonstrated and established their key role in both the generation of precancerous lesions and their progression to invasiveness, as well as their multipronged roles in the acquirement of many malignancy hallmark features, such as sustained proliferation, insensitivity to proapoptotic signals and the acquisition of migratory and invasive capacity [16,17]. Among the most prominent functions of the Hippo pathway is its ability to orchestrate cell interactions with various therapeutic regimens, including chemotherapy–radiotherapy and immunotherapy. In this context, it is capable of exerting a tumor-suppressive role after neoplastic cell exposure to chemotherapeutic agents or radiation, which invoke damage to their DNA via promoting DNA-damage-related programmed cell death and restricting the survival and outgrowth of genomically unstable cells. LATS2 is activated via phosphorylation by Chk1 and then induces activation of ATR and phosphorylation of ASSP1, facilitating the formation of multiprotein complex LATS2/ASSP1/p53, which then translocates to the cell nucleus, stimulating transcription of apoptosis-related genes. LATS2 also interacts with mouse double minute 2 (MDM2) protein and suppresses its E3 ubiquitin ligase activity towards p53, enhancing the stability and prolonging the half-life of the latter. Treatment with the spindle-damaging agent nocodazole leads to LATS2 nuclear translocation. On the other hand, in response to DNA damage, LATS1 is implicated in the regulation of G2/M transition by inactivating Polo-like kinase 1 (Plk1), a serine/threonine-protein kinase, that progresses the cell-cycle and entry into the mitotic phase. Another intriguing function of YAP and TAZ, of paramount translational/clinical interest and with profound implications in routine patient treatment, is the upregulation of PD-L1, which designates the eminent impact of YAP and TAZ in regulating tumor cell immunogenicity and has been recorded in heterogenous malignancies [43,44,45,46,47,48,49,50,51]. 

Persistent activation of YAP and TAZ is mediated by numerous paths, which can be classified into two mutually exclusive categories based on whether the deregulation is detected in YAP and TAZ themselves or in upstream components of the Hippo pathway. The two main executors of the cascade are frequently overexpressed in various malignancies via amplification of the genes’ loci or fusions with other genes [52,53,54,55]. The second group of alterations identified as drivers of Hippo constant activation includes aberrant activity of upstream signal transducers, such as inactivation of NF2 or LATS1/2 via mutations or hypermethylation [53,56]. In thymic tumors, the only identified mechanism implicated in YAP deregulation is a chromosomal rearrangement that leads to the generation of a hybrid *YAP-MAML2* gene. This gene fusion, however, pertains only to a tiny fraction of TETs termed at metaplastic thymomas, which are characterized by benign histological features, indolent behavior and favorable prognosis [57,58]. 

Some intriguing data from our study concern the subcellular localization of YAP, TAZ and TEAD4, as their distribution among different cell compartments directly determines both their transcriptional impact and their possible non-transcriptional roles. The recently emerged data regarding the multidimensional role of YAP/TAZ and TEAD4 in cellular physiology, confront the well-established on-off theory, which postulates that they are inactive when preserved in the cytoplasm, as they seem to get involved in crosstalk with complementary pathways, enabling them to perform additional functions. While classical models initially considered the nucleocytoplasmic translocation of YAP/TAZ as a binary state, recent data have clarified the intricate regulatory networks that act simultaneously on the two factors and, through multiple different import and export signals, create a dynamic model of continuous shuttling between nuclear and cytoplasmic compartments [59]. These complex signals affecting YAP/TAZ nuclear influx/efflux rates are both passive, such as regulation of membrane permeability, and active, which include post-translational modifications and interactions with other proteins [59]. The latter process involves crosstalk with other complementary pathways, such as Src or 14-3-3 [60,61,62]. The continuous nucleocytoplasmic dynamics driven by these heterologous cellular signals create a fluctuating equilibrium of two contradictory mechanisms and the strength of each respective signal determines the direction to which the balance tilts and, subsequently, the exact subcellular localization of the two factors at each moment. Despite the multifarious intracellular networks which impact the YAP/TAZ localization and activation state, it is still believed that upstream components of the Hippo pathway are, arguably, the main determinants of their activity. The binding of Hippo ligands to their receptors leads to an intracellular phosphorylation cascade of core components, namely Macrophage Stimulating 1/2 (MST1/2) and LATS1/2, ending up with the addition of phosphate groups to YAP/TAZ. This first phosphorylation seems to facilitate conformational modifications of the two effector molecules, rendering them prone to additional phosphorylation events by 14-3-3 factor and, consequently, to their cytoplasmic retention. The Wnt pathway is also implicated in YAP/TAZ regulation, as it has been shown to stimulate TAZ stabilization and cytoplasmic retention. Furthermore, this function implies the existence of a negative feedback loop, as cytoplasmic YAP and TAZ modulate β-catenin activity via direct or indirect ways. Cytoplasmic YAP may directly shuttle β-catenin into the cytoplasm, while cytoplasmic TAZ prompts β-catenin accumulation via sequestering DVL2 and impairing its activity [63,64,65]. TEAD4, which is considered as the final determinant of Hippo pathway transcriptional output, also seems to be subject to cytoplasmic sequestration, especially in stress conditions. This tendency of such a conventional transcription factor to translocate from its natural space of action to the cytoplasm seems to be completely Hippo-independent with the stress-induced p38 MAPK pathway considered to be the main driver of its nuclear export [66]. Indeed, p38 directly interacts with TEAD4, promoting its shift to the cytoplasm via a Chromosomal Maintenance 1 (CRM1)-dependent process and not through direct phosphorylation. The dependence of TEAD4 cytoplasmic translocation solely on p38 was also confirmed via pharmacological inhibition of both p38 and Hippo, with the first abrogating TEAD4 nuclear export and the latter failing to alter its distribution [66]. TEAD4 has been detected in the cytoplasm in the early stages of embryonic development, more specifically during initial lineage specification in blastocyst stage [67]. While cytoplasmic shuttling apparently has an inhibitory effect on its transcription function, the exact biological importance and the functions of TEAD4 in the cytoplasm remain a completely black box. Regarding LATS1, given its function as a negative regulator of YAP/TAZ, its low cytoplasmic level harmonizes with the high activity of the downstream cascade components. However, besides its primary cytoplasmic kinase function, LATS1 has also been shown to possess the capability of nuclear translocation, triggered by Interferon-γ (IFNγ). In the nucleus, it retains its kinase activity by adding phosphate groups to STAT1 [68], modifying immune-related pathways.

The growing evidence supporting the multipronged role of the Hippo pathway in cancer pathogenesis has fueled a progressively growing interest in the development of strategies that will enable us to therapeutically interfere with its activity and impede its protumoral potential. YAP and TAZ, along with TEADs, are considered the most “druggable” sites of the Hippo pathway. Potential therapeutic interventions can either target the interactions among them or inhibit the activity of distinct factors. The main representative of the first category are molecules targeting the sites of interaction between YAP/TAZ and TEAD, which bear a highly conserved palmitoylation pocket, vulnerable to blockade by small molecules, such as verteporfin [69,70,71]. Verteporfin is a porphyrin derivative that draws its pharmacological activity from the generation of reactive oxygen radicals and has been approved by the FDA for the treatment of wet age-related macular degeneration (AMD) [72]. Verteporfin treatment has been shown to inhibit the expression of Hippo pathway target genes, while it is also capable of reorganizing the structure of the cytoskeleton, a process implicated in the regulation of different aspects of cellular physiology [72,73,74]. Numerous other compounds intervening in YAP/TEAD interaction include compound 5, which was shown to inhibit target gene expression and cell proliferation in a liver cell line, HuH7 and flufenamic acid derivatives (compound 7), composed of chloromethyl ketone moieties that recognize and bind to conserved cysteine motifs in the lipid pocket, impairing YAP–TEAD interaction and transcriptional activity [75,76]. Another family of drugs belonging to the same category are cyclic YAP-like peptides, which also share conformational similarities and interfere with endogenous YAP binding to GST-TEAD1 in cell lysate [77]. Finally, a synthetic peptide aims to take advantage of another family of transcriptional co-factors of YAP/TEAD, namely mimicking VGLL4 proteins called Vestigial-like protein (VGLL) 1–4 structure, known as ‘super-TDU’ is able to interfere with TEAD binding to YAP [78]. The second category of inhibitors engulfs molecules that target either YAP or TEAD. The prototypical representative of this class is CA3, which has been reported to inhibit YAP by reducing its expression [79]. Following its development, a number of synthetic compounds, sharing structural similarities with CA3, have been manufactured, showing great promise in suppressing YAP activity [76]. Given the fact that the majority of available synthetic compounds interfering with Hippo activity hinder YAP transcriptional functions, it seems that type A thymomas, which demonstrated the highest nuclear YAP compared to the remaining histological categories, could be more prone to these types of treatments.

Our findings clearly represent some preliminary promising data concerning the aptitude of Hippo pathway components as biomarkers, helping in the precise selection of patients who will benefit from emerging targeted therapies. Especially, the previously delineated interrelation of the Hippo pathway with PD-L1 expression level combined with the continuous activity in the field of pharmaceutic industry for direct Hippo therapeutic suppression creates a very promising field of research in the direction of generation of promising combination targeted therapeutic schemes, whose implementation can add even more options in the battle against cancer.

On the other hand, our findings come with certain limitations, which makes it essential that they be verified by additional studies. First of all, our estimation of the four Hippo components expression was based solely on immunohistochemistry and this, by itself, makes it necessary that our findings be interpreted with a grain of salt. Moreover, while the 88 cases included in our analysis are not a negligible amount, the substantial histological heterogeneity of this tumor family means that the representation of each distinct histological subtype is restricted, limiting the generation of more precise statistical correlations. In addition, regarding the clinical information (neoadjuvant treatment, survival, development of Myasthenia Gravis), we were able to obtain them only for a relatively small fraction of our patients, which imposed severe restrictions on the unraveling of the statistical correlations between the expression patterns of the examined proteins and these clinical parameters. Finally, in the era of molecular biology revolution, which has taken place in the last decades, immunohistochemical detection of a protein should be followed by more in-depth molecular techniques, which can provide a more explicit quantification of its expression levels, along with other important information, such as its transcriptional rate.

## 4. Materials and Methods

### 4.1. Patients

Our cohort consisted of 88 patients from the Evangelismos General Hospital in Athens, Greece, with TETs from a period of 10 years (2009–2019). All patients had undergone surgical resection and their formalin-fixed paraffin-embedded (FFPE) tissue and medical records were available. The study was conducted in accordance with the Declaration of Helsinki and it was approved by the Bioethics Committee of the National and Kapodistrian University of Athens, Greece (protocol code 140/27 June 2019). Patient characteristics are shown in Table 1. Thirty-nine of the patients were men (44%) and forty-nine were women (56%), with a median age at diagnosis of 62 years (range 27–88 years). The frequency of WHO subtypes according to the latest WHO classification [80] was as follows: type A, 12.5%; type AB, 20.4%; type B1, 15.9%; type B2, 21.6%; type B3, 15.9%; micronodular thymoma with lymphoid stroma (MNT), 2.3%; and thymic carcinoma (TC), 11.4%. Masaoka–Koga stage was I in 16.5%; IIa in 39.2%; IIb in 16.5%; III in 20.2%; IVa in 3.8%; and IVb in 3.8% of patients. Chemotherapy and radiotherapy were administered to 26.3% and 50%, respectively, of the patients for whom respective information was available; 6 of these patients received both. Follow-up information was available for 40 patients, with a median period of 32 months (range: 7–134 months).

### 4.2. TMA Construction

We created tissue microarrays (TMAs) by selecting representative areas from one tissue block of each of the neoplasms of our cohort. For the construction of TMAs, we used a manual tissue arrayer (TMA Model I, Beecher Instruments, Sun Prairie, WI, USA). Three to five 1.5 mm cores of the selected areas were transferred from their respective FFPE block to the recipient paraffin blocks in order to capture intratumoral heterogeneity.

### 4.3. Immunohistochemistry

Immunohistochemical stains were conducted with antibodies against YAP (abcam rabbit polyclonal ab114862/dilution 1:100), TAZ (abcam mouse monoclonal ab242313/dilution 1:50), TEAD4 (abcam rabbit polyclonal ab97460/dilution 1:100) and LATS1 (abcam rabbit polyclonal ab111344/dilution 1:50). Staining of samples with each antibody was carried out based on the respective protocol referred to on its datasheet. As positive control for all four antibodies, we used placental tissue. Immunohistochemical assessment was based on the estimation of H-score, a semiquantitative measure of protein levels. Calculation of H-score is based on multiplying the semiquantitative staining intensity (score 1 to 3) by the percentage of positive cells. As a result, H-score ranges between 0 and 300. The epithelial neoplastic cells and the lymphocytes, as well as the nuclear and cytoplasmic staining, were separately assessed. Estimation of H-score for each sample was conducted by two independent pathologists (S.T. and K.P.) with complete interobserver compliance.

### 4.4. Statistical Analysis

Statistical analysis was carried out by a MSc biostatistician (GL). The correlation between the nuclear and cytoplasmic H-scores of YAP, TAZ, TEAD4 and LATS1 with clinicopathological parameters was evaluated with nonparametric tests. Correction for multiple comparisons was applied when necessary. For survival analysis, Kaplan–Meier curves were generated and the log-rank test was used to compare their differences. A cut-off of 0.05 was used for statistical significance. The analysis was conducted with the statistical package STATA 11.0/SE (College Station, TX, USA) for Windows.

## 5. Conclusions

In this study, we provide the first comprehensive evaluation of four Hippo pathway components’ (YAP, TAZ, TEAD4 and LATS1) expression pattern in a cohort of TETs, with interesting results regarding their levels and distribution within tumor cells. While increased nuclear YAP and TAZ are correlated with less aggressive tumors, higher cytoplasmic TAZ, TEAD4 and LATS1 are associated with high-grade and more advanced stage neoplasms. On the other hand, no significant association was revealed between the examined factors and the patients’ prognoses. Our results provide some initial insight regarding the suitability of these factors as future biomarkers, which can be gradually incorporated in everyday clinical practice.

## Figures and Tables

**Figure 1 biomedicines-11-01876-f001:**
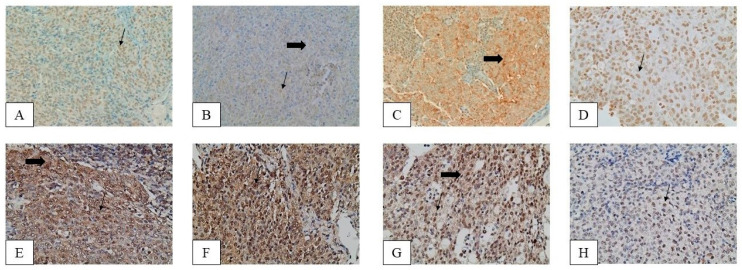
Immunohistochemical expression of YAP, TAZ, TEAD4 and LATS1 in the epithelial component of TETs. YAP in two A-type thymomas [(**A**) nuclear (H-score: 140), (**B**) nuclear (H-score: 25) and cytoplasmic (H-score: 80)]. TAZ in a type B3 thymoma [(**C**) cytoplasmic (H-score: 300)] and in a thymic carcinoma [(**D**) nuclear (H-score: 180)]. TEAD4 in a thymic carcinoma [(**E**) nuclear (H-score: 160) and cytoplasmic (H-score: 160)] and in a type B3 thymoma [(**F**) nuclear (H-score: 270) and cytoplasmic (H-score: 200)]. LATS1 in a type B3 thymoma (**G**) [nuclear (H-score: 180) and cytoplasmic (H-score: 90)] and in a type B2 thymoma [(**H**) nuclear (H-score: 120)]. All pictures are at ×400 magnification. All thin arrows show indicative areas of nuclear staining and thick arrows show areas of cytoplasmic staining.

**Figure 2 biomedicines-11-01876-f002:**
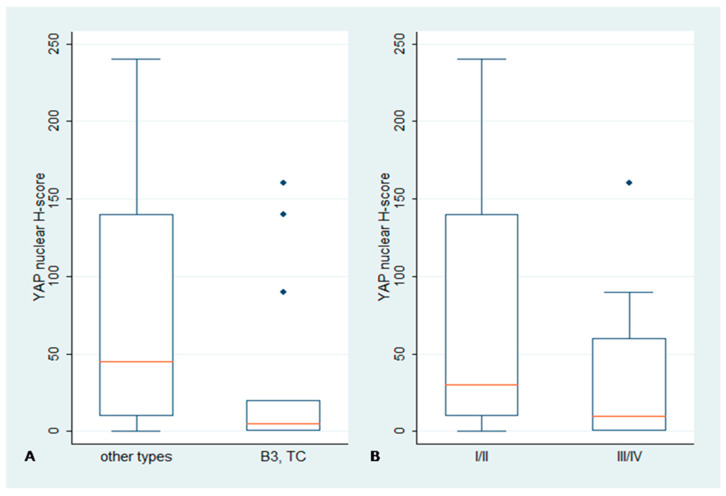
Association between nuclear YAP H-score and (**A**) WHO type or (**B**) Masaoka–Koga stage. The red line denotes median value. Lower nuclear H-score was observed in epithelial-rich TETs (B3, TC) compared to the other TETs (*p* = 0.0042) (**A**). Moreover, advanced Masaoka–Koga stages (III/IV) were associated with a lower nuclear H-score (*p* = 0.042) (**B**).

**Figure 3 biomedicines-11-01876-f003:**
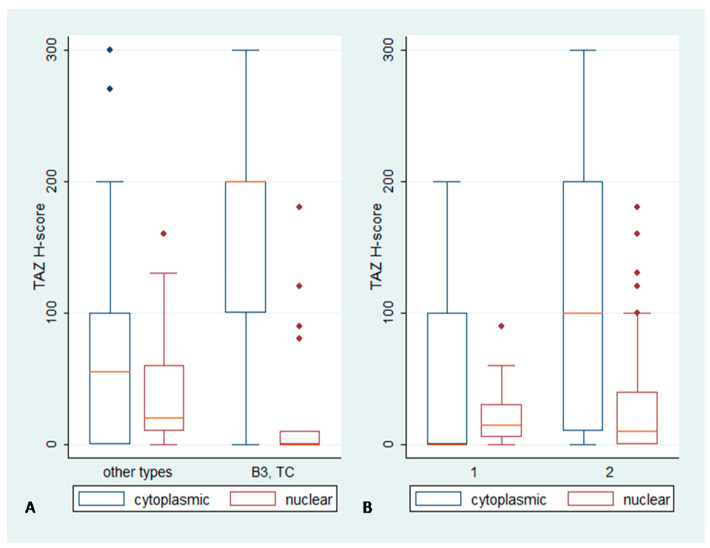
Association between cytoplasmic and nuclear TAZ H-score and (**A**) WHO type or (**B**) Masaoka–Koga stage. The red line denotes median value. Higher cytoplasmic and lower nuclear H-score was observed in epithelial-rich TETs (B3, TC) compared to the other types (*p* = 0.0001 for cytoplasmic and *p* = 0.0215 for nuclear) (**A**). Advanced Masaoka–Koga stages (III/IV) were associated with higher cytoplasmic H-score (*p* = 0.0308) (**B**).

**Figure 4 biomedicines-11-01876-f004:**
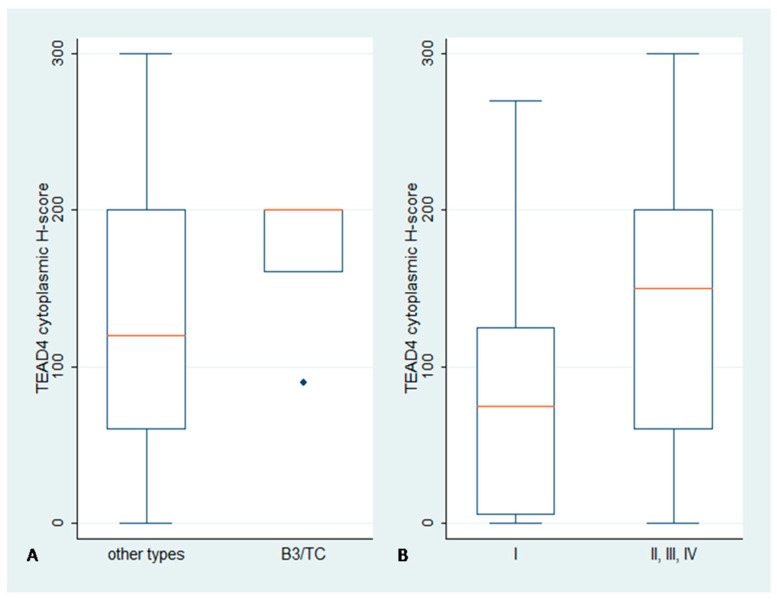
Association between TEAD4 cytoplasmic H-score and (**A**) WHO type or (**B**) Masaoka–Koga stage. The red line denotes median value. Higher cytoplasmic H-score was observed in epithelial-rich TETs (B3, TC) compared to the other TET types (*p* = 0.002) (**A**). Moreover, advanced Masaoka–Koga stages (II/III/IV) were associated with a marginally higher cytoplasmic H-score (*p* = 0.05) (**B**).

**Figure 5 biomedicines-11-01876-f005:**
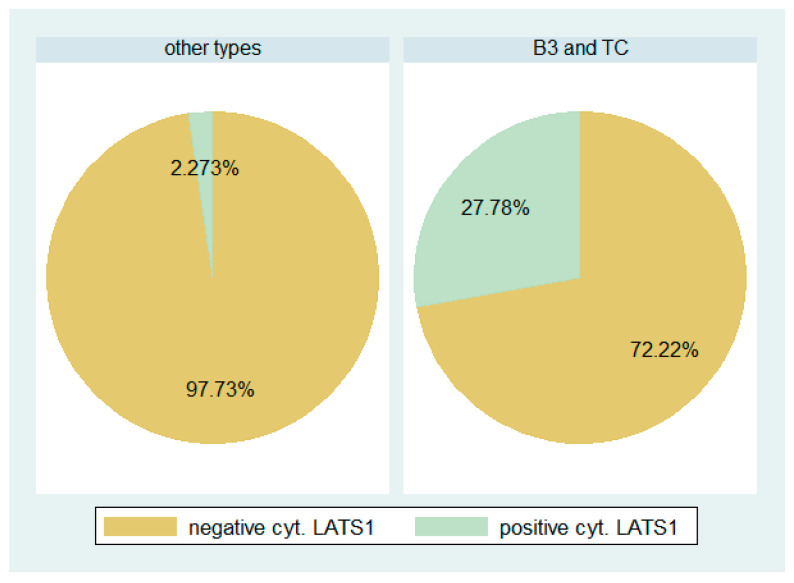
Cytoplasmic LATS1 expression between B3 and TC compared to other TETs. The epithelial-rich TETs (B3, TC) more often displayed a cytoplasmic staining compared to the other TETs (*p* = 0.006). Moreover, advanced stage tumors (III/IV) more often displayed cytoplasmic positivity (*p* = 0.04).

**Table 1 biomedicines-11-01876-t001:** Clinicopathological characteristics of 88 patients with TETs.

Parameter	Median	Range
Age	62	27–88 years
	**Number**	**%**
Gender		
Male	39/88	44%
Female	49/88	56%
WHO subtypes		
Type A	11/88	12.5%
Type AB	18/88	20.4%
Type B1	14/88	15.9%
Type B2	19/88	21.6%
Type B3	14/88	15.9%
Micronodular with lymphoid stroma	2/88	2.3%
Thymic Carcinoma	10/88	11.4
Masaoka–Koga stage		
I	13/79	16.5%
IIa	31/79	39.2%
Iib	13/79	16.5%
III	16/79	20.2%
Iva	3/79	3.8%
Ivb	3/79	3.8%
Presence of Myasthenia Gravis	35/58	60.3%
Presence of chemotherapy	10/38	26.3%
Presence of radiotherapy	19/38	50%
Event		
Alive	29/40, follow-up 5–134 months	72.5%
Dead of disease	11/40, within 7–65 months	27.5%
Presence of relapse	4/35, within 58–65 months	11%

**Table 2 biomedicines-11-01876-t002:** Expression of YAP, TAZ, TEAD4 and LATS1 in TETs.

Epithelial Cells	Positivity Rate	H-Score, Median	H-Score, Range
YAP nuclear expression	76.9%	20	0–240
YAP cytoplasmic expression	32%	0	0–300
TAZ nuclear expression	74.7%	10	0–180
TAZ cytoplasmic expression	72.1%	90	0–300
TEAD4 nuclear expression	98.3%	160	0–300
TEAD4 cytoplasmic expression	91.4%	140	0–300
LATS1 nuclear expression	77.4%	80	0–300
LATS1 cytoplasmic expression	9.7%	0	0–120

## Data Availability

The data presented in this study are available on request from the corresponding author.

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
