# Peer review of "Searching for Novel Biomarkers in Thymic Epithelial Tumors: Immunohistochemical Evaluation of Hippo Pathway Components in a Cohort of Thymic Epithelial Tumors"

_biomedicines, 2023, doi:10.3390/biomedicines11071876_

Round 1

Reviewer 1 Report

In this study, the authors tried to evaluate the expression patters of YAP and TAZ in a cohort of thymic epithelial tumors (TETs), distributed across the whole breadth of their histological subclasses and correlate them with available clinicopathological parameters. The authors concluded that “Our findings clearly represent some preliminary promising data concerning the aptitude of Hippo pathway components as biomarkers, helping in the precise selection of patients who will benefit from emerging targeted therapies”.

Comments:

The reviewer has some concerns as follows:

1. In the introduction, the description of the research motivation seems insufficient. Is there any literature or previous studies or big data prediction system to support the possible role of Hippo pathway in TETs?

2. In terms of methods, only tissue immunohistochemical method was used to explain the expression of proteins for Hippo pathway, and the evidence is insufficient.

3. How to quantify the levels of protein expression in IHC? The detailed description should be shown in the Methods.

4. In Figure 1, the IHC images are unconvincing. Where is the cytoplasmic expression or the nuclear expression? Moreover, the scale bars are lacking.

5. In Figures 2-5, the descriptions in the legends are too simple to understand the presentation of the results. The experimental conditions including n numbers can be described in the legends and the statistical significance can be shown into the Figures.

6. In general, this study is a very preliminary study, and the evidence for this Hippo pathway as a biomarker is insufficient.

Author Response

We would like to thank the reviewer for his meaningful comments. Please, find our responses below:

  1. In the introduction, the description of the research motivation seems insufficient. Is there any literature or previous studies or big data prediction system to support the possible role of Hippo pathway in TETs?

There are no previous studies, describing the role of Hippo pathway components in thymic epithelial tumors (TETs). The only findings, indicating a role of Hippo pathway in thymic tumorigenesis have been derived from a molecular analysis, of a TETs cohort, which revealed an upregulation of YAP in tumors of A subtype and from the study of metaplastic thymomas which bear a MAML2-YAP translocation. These findings are described in the “Discussion” section, in lines 187-190 and 217-221 respectively. However, the main motivation for our study was an in vivo analysis of the role of YAP and TAZ in thymic development, which implied an important role of both factors in its normal development and homeostasis (newly added reference 5 in line 49).

  1. In terms of methods, only tissue immunohistochemical method was used to explain the expression of proteins for Hippo pathway, and the evidence is insufficient.

We have added a paragraph in the end of “Discussion” section, describing the limitations of the study, where we refer among others the fact that additional molecular studies are needed, so that the role of Hippo pathway in thymic tumorigenesis is more clearly elucidated.

  1. How to quantify the levels of protein expression in IHC? The detailed description should be shown in the Methods.

The method used to estimate H-score is described in “Materials and Methods” section, in lines 346-352.

  1. In Figure 1, the IHC images are unconvincing. Where is the cytoplasmic expression or the nuclear expression? Moreover, the scale bars are lacking.

We have added arrows in each of the six pictures (A-F), to indicate the cytoplasmic and nuclear staining in each case.

  1. In Figures 2-5, the descriptions in the legends are too simple to understand the presentation of the results. The experimental conditions including n numbers can be described in the legends and the statistical significance can be shown into the Figures.

We have added at the legends of figures 2, 3, 4 and 5 more detailed description of the results, along with the p-values, when statistical significance existed.

  1. In general, this study is a very preliminary study, and the evidence for this Hippo pathway as a biomarker is insufficient.

We have added a paragraph in the end of “Discussion” section, describing the limitations of the study, where we refer among others the fact that additional molecular studies are needed, so that the role of Hippo pathway in thymic tumorigenesis is more clearly elucidated.

Reviewer 2 Report

Dear Authors,

An interesting manuscript with some places to be improved:

1) Introduction.Line 36 -  you mentioned cells involved in tumorogenesis. could you please more specify what type of thymus cells do you mean, which are most responsible for the tumor development;

2) decipher please all the abbreviations, - you have to write full title if you mention the term for the first time!

3) move, please the Table 1 for the material and Methods section;

Materials and methods.

4.1 Put here please the Ethical Committee permission - issue office, code, date. Specify the patient inclusion/exclusion criteria

4.3 Please for each IMH antibody - add the working dilution, company, from which the antibody was obtained. Additionally, give please reference to the IMH protocol or give detailed its description, including the secondary kit used; Decipher (describe) the semi-quantitative evaluation, give also reference to it. Add please a Supplementary File with example of microphotos for each staining semi-quantitative score. I couldnt find description of positive and negative controls for each antibody, please, add it! Finally, I believe that evaluation was done at least for two independent morphologists, - it should be mentioned here then!

Results. Fig. 4. Indicate (describe) the semi-quantitative score here for each microphoto in description.

Discussion. Lacks the Limitation paragraph at the end.

Conclusions. Dear Authors, this is not a section of brief story! Please, remove the extra sentences, words, shorten them and make them more precise with explanation of obtained data.

References.

33 - please decipher this one!

Author Response

We would like to thank the reviewer for his meaningful comments. Please, find our responses below:

1) Introduction.Line 36 -  you mentioned cells involved in tumorogenesis. could you please more specify what type of thymus cells do you mean, which are most responsible for the tumor development;

We added a sentence (lines 36-39) about the cells that are believed to give origin to thymic epithelial tumors.

2) decipher please all the abbreviations, - you have to write full title if you mention the term for the first time!

We have deciphered abbreviations which had not been written in their full form in the text: TEAD4 (line 65), LATS1/2 (line 55), MST1/2 (line 262-263) and TC (86-87).

3) move, please the Table 1 for the material and Methods section;

Table 1 is after “Introduction” because is cited in the last sentence of “Introduction”. Moreover, it needs to be placed before the other figures, as it includes the basic clinicopathological data regarding the population of our study.

Materials and methods.

4.1 Put here please the Ethical Committee permission - issue office, code, date. Specify the patient inclusion/exclusion criteria

We have added the information regarding the Ethical Committee permission - issue office, code and date at the “Materials and methods” section (lines 319-321). Regarding the criteria of patients inclusion/exclusion, we included in the study all the patients from Evangelismos Hospital, for which there was available pathology report, as specified in “Materials and methods” section, in lines 316-319.

4.3 Please for each IMH antibody - add the working dilution, company, from which the antibody was obtained. Additionally, give please reference to the IMH protocol or give detailed its description, including the secondary kit used; Decipher (describe) the semi-quantitative evaluation, give also reference to it. Add please a Supplementary File with example of microphotos for each staining semi-quantitative score. I couldnt find description of positive and negative controls for each antibody, please, add it! Finally, I believe that evaluation was done at least for two independent morphologists, - it should be mentioned here then!

The information of antibodies company, with the exact product number are included in the “Materials and Methods” section (lines 341-344). Moreover, we added the dilution used for each antibody. Regarding the exact protocols used, as referred in the same paragraph (lines 344-345), they are available at the datasheet of each antibody. The method used to estimate H-score is also described in “Materials and Methods” section, in lines 346-352. Concerning positive and negative controls, we used placental tissue as positive control for all antibodies, as referred in line 345, while negative control was not used, as no specific negative control was suggested at the datasheets of the antibodies. We have also added a mention that the evaluation of immunohistochemical stainings was conducted by two independent pathologists (lines 352-354). Finally, instead of adding a supplementary file, we have added the H-scores of the photos at figure 1.

Results. Fig. 4. Indicate (describe) the semi-quantitative score here for each microphoto in description.

We have added the H-scores of the cases, corresponding to the six photos in figure 1.

Discussion. Lacks the Limitation paragraph at the end.

We have added a paragraph at the end of the “Discussion” regarding the limitations of our study (lines 348-362).

Conclusions. Dear Authors, this is not a section of brief story! Please, remove the extra sentences, words, shorten them and make them more precise with explanation of obtained data.

References.

33 - please decipher this one!

We have deciphered the citation.

Reviewer 3 Report

A study by Palamaris et al investigates novel biomarkers and therapeutic targets in thymic epithelial tumors focusing on the Hippo pathway. The study is a little bit preliminary, althought he Authors used patient-derived samples. The Authors provided observations that can drive further hypotheses that need to be validated experimentally. As the Authors state "Despite the indisputable limits characterizing studies based solely on immunohistochemical detection of proteins...". To substantiate their work, the Authors should consider to analyze publicly available data e.g., RNA-seq with regard to expression of Hippo pathway-associated genes in thymic epithelial tumors. This might improve the significance of this study. The manuscript is clearly written in a good English.

Specific comments:

1. I recommend to change the title as it suggests that this is a review paper.

Author Response

 We would like to thank the reviewer for his meaningful comments. Please, find our responses below:

A study by Palamaris et al investigates novel biomarkers and therapeutic targets in thymic epithelial tumors focusing on the Hippo pathway. The study is a little bit preliminary, althought he Authors used patient-derived samples. The Authors provided observations that can drive further hypotheses that need to be validated experimentally. As the Authors state "Despite the indisputable limits characterizing studies based solely on immunohistochemical detection of proteins...". To substantiate their work, the Authors should consider to analyze publicly available data e.g., RNA-seq with regard to expression of Hippo pathway-associated genes in thymic epithelial tumors. This might improve the significance of this study. The manuscript is clearly written in a good English.

Specific comments:

  1. I recommend to change the title as it suggests that this is a review paper.

We have changed the manuscript title.

Reviewer 4 Report

The manuscript represents an interesting analysis of members of the Hypo pathway in thyroid tumours. The manuscript, in general, is well-written, and the general statistics are adequate. There are only minor details which may enhance the manuscript. One of them is the relation between treatment and the up or down-modulation in the expression of the antigens. Usually, chemotherapy and radiotherapy induce the expression of stress proteins which may involve modulation of the Hypo pathway. The authors should discuss this point. Finally, minor details in the text should be corrected. For example, table 1 shows the presence of Myasthenia gravis for only Myasthenia gravis and so forth. The four outliers in Figure 3 should be properly discussed. The manuscript will benefit from a section on limitations encountered.

The language is appropiate

Author Response

We would like to thank the reviewer for his meaningful comments. Please, find our responses below:

The manuscript represents an interesting analysis of members of the Hypo pathway in thyroid tumours. The manuscript, in general, is well-written, and the general statistics are adequate. There are only minor details which may enhance the manuscript. One of them is the relation between treatment and the up or down-modulation in the expression of the antigens. Usually, chemotherapy and radiotherapy induce the expression of stress proteins which may involve modulation of the Hypo pathway. The authors should discuss this point. Finally, minor details in the text should be corrected. For example, table 1 shows the presence of Myasthenia gravis for only Myasthenia gravis and so forth. The four outliers in Figure 3 should be properly discussed. The manuscript will benefit from a section on limitations encountered.

We have added a brief paragraph, describing the interactions of Hippo pathway with stress proteins, induced after chemotherapy-radiotherapy in the “Discussion” section (lines 223-238).

Moreover, we added a paragraph referring to the study limitations, in the “Discussion” section (line 348-362).

Round 2

Reviewer 1 Report

This revised manuscript has been greatly improved. The authors revised the title and text. The reviewer has no further comments.

Reviewer 2 Report

Dear Authors,

Well, not bad. However, I am slightly upset about the missed SUPPL file with the scores. I understand the morphology, but not all the readers will be satisfied with the H-scores as their variations are very high in your pictures! But Ok, - as scientist you have right for your own opinion and thus I advice to publish your manuscript!